# Associations of Changes in Blood Lipid Concentrations with Changes in Dietary Cholesterol Intake in the Context of a Healthy Low-Carbohydrate Weight Loss Diet: A Secondary Analysis of the DIETFITS Trial

**DOI:** 10.3390/nu13061935

**Published:** 2021-06-04

**Authors:** Monica Vergara, Michelle E. Hauser, Lucia Aronica, Joseph Rigdon, Priya Fielding-Singh, Cynthia W. Shih, Christopher D. Gardner

**Affiliations:** 1Department of Health Research & Policy, Stanford University School of Medicine, Stanford, CA 94305, USA; mvergara@alumni.stanford.edu; 2General Surgery, Department of Surgery, Stanford University School of Medicine, Stanford, CA 94305, USA; mehauser@stanford.edu; 3Medical Service-Obesity Medicine, Veterans Affairs Palo Alto Health Care System, Palo Alto, CA 94304, USA; 4Internal Medicine-Primary Care, Fair Oaks Health Center, San Mateo County Health System, Redwood City, CA 94063, USA; 5Stanford Prevention Research Center, Stanford University School of Medicine, Stanford, CA 94305, USA; laronica@stanford.edu; 6Quantitative Sciences Unit, Stanford University School of Medicine, Stanford, CA 94305, USA; jrigdon@wakehealth.edu (J.R.); cshih@alumni.stanford.edu (C.W.S.); 7Huntsman Cancer Institute, University of Utah, Salt Lake City, UT 84112, USA; priya.fielding-singh@hci.utah.edu

**Keywords:** dietary cholesterol, low-carbohydrate, LDL cholesterol, HDL cholesterol, triglycerides, human study, weight loss trial, healthy adults, diet quality, eggs

## Abstract

In 2015, the Dietary Guidelines for Americans (DGA) eliminated the historical upper limit of 300 mg of dietary cholesterol/day and shifted to a more general recommendation that cholesterol intake should be limited. The primary aim of this secondary analysis of the Diet Intervention Examining the Factors Interacting With Treatment Success (DIETFITS) weight loss diet trial was to evaluate the associations between 12-month changes in dietary cholesterol intake (mg/day) and changes in plasma lipids, particularly low-density lipoprotein (LDL) cholesterol for those following a healthy low-carbohydrate (HLC) diet. Secondary aims included examining high-density lipoprotein (HDL) cholesterol and triglycerides and changes in refined grains and added sugars. The DIETFITS trial randomized 609 healthy adults aged 18–50 years with body mass indices of 28–40 kg/m^2^ to an HLC or healthy low-fat (HLF) diet for 12 months. Linear regressions examined the association between 12-month change in dietary cholesterol intake and plasma lipids in 208 HLC participants with complete diet and lipid data, adjusting for potential confounding variables. Baseline dietary cholesterol intake was 322 ± 173 (mean ± SD). At 12 months, participants consumed an average of 460 ± 227 mg/day of dietary cholesterol; 76% consumed over the previously recommended limit of 300 mg/day. Twelve-month changes in cholesterol intake were not significantly associated with 12-month changes in LDL-C, HDL-C, or triglycerides. Diet recall data suggested participants’ increase in dietary cholesterol was partly due to replacing refined grains and sugars with eggs. An increase in daily dietary cholesterol intake to levels substantially above the previous 300 mg upper limit was not associated with a negative impact on lipid profiles in the setting of a healthy, low-carbohydrate weight loss diet.

## 1. Introduction

Poor diet is a key risk factor for cardiovascular disease (CVD), the leading cause of death in the United States [1]. Until recently, a “heart-healthy diet” included a recommendation to limit dietary cholesterol to no more than 300 mg/day [2,3,4,5]. This recommendation was based on a body of evidence that included both interventional studies and observational epidemiology studies suggesting links between dietary cholesterol and blood concentrations of LDL cholesterol (LDL-C), a major risk factor for CVD [6,7,8]. However, evidence has failed to show a clear dose–response relationship between dietary cholesterol intake and LDL-C levels [9,10,11]. A lack of clear evidence for selecting 300 mg/day of dietary cholesterol as a specific upper limit led to the removal of the upper limit in the 2015–2020 Dietary Guidelines for Americans (DGA), although there is still a general recommendation to limit cholesterol intake [12].

A 2019 American Heart Association (AHA) Science Advisory position statement reviewed the existing evidence on the associations between dietary cholesterol, LDL-C, and CVD with the aim of providing recommendations to clinicians [13]. Discrepancies between different studies’ findings were examined and highlighted the importance of considering two important features: (1) confounding with saturated fat, which is found in most foods high in cholesterol, and (2) overall diet quality and food patterns, which may modify the effect of cholesterol on LDL-C and CVD risk [14]. Eggs, in particular, have been singled out in debates about the health impacts of dietary cholesterol as they are the most consumed high-cholesterol food in the United States yet have relatively low levels of saturated fat. A recent prospective study of a half-million people in nine European countries reported an inverse relationship between egg intake and ischemic heart disease [15]. There is a paucity of data on the effects of a high-cholesterol diet or a diet high in egg consumption where this intake occurs in the context of an overall high-quality diet.

The Diet Intervention Examining the Factors Interacting with Treatment Success (DIETFITS) randomized controlled weight loss trial provides an opportunity to examine the effect of increased dietary cholesterol intake on blood concentrations of LDL-C in the context of consuming a high-quality diet. In addition to focusing on weight loss, the intervention aimed to help participants in one arm reduce carbohydrate intake via a healthy low-carbohydrate (HLC) diet. Many of the low-carbohydrate foods regularly chosen by participants in this arm were animal-derived, particularly eggs, and were therefore high in cholesterol. Extensive data collection over a 12-month period (i.e., baseline, 3, 6, and 12 months) included plasma lipid measurements and multiple 24-hour diet recalls [16,17]. In this secondary analysis of the DIETFITS trial, the primary objective was to assess the associations between 12-month change in dietary cholesterol intake and 12-month change in plasma lipids, particularly LDL-C. Secondary objectives included exploration of the extent to which changes in dietary cholesterol intake were due to changes in egg consumption and whether changes in egg consumption were associated with changes in refined grain and added sugar intake.

## 2. Materials and Methods

### 2.1. Study Design

Detailed methods of the DIETFITS trial were published previously [16]. In brief, it was a randomized controlled weight loss trial of 609 participants assigned to follow either HLC or healthy low-fat (HLF) diets. The original trial employed a parallel study design with the primary aim of assessing the effects of both diets on weight change and whether these effects were modified by genetic factors (i.e., multilocus genotype pattern) or insulin secretion over a 12-month period. The original randomized controlled trial was powered to detect this effect modification, with a sample size of around 300 per arm. Participants were instructed to focus on eating high-quality diets based on whole foods while achieving the lowest intake of either net carbohydrate (HLC) or fat (HLF) they could maintain long-term; there was no specific caloric restriction target. Informed written consent was obtained from all participants. 

Participants were men and premenopausal women aged 18 to 50 years with body mass indices (BMI, kg/m^2^) of 28 to 40, who were otherwise generally healthy. Exclusion criteria included pregnancy, lactation, being within 6 months postpartum, or planning to become pregnant during the study; diabetes, metabolic disorders, or cancer; cardiovascular, renal, or liver disease; or use of any medications that altered lipids, blood pressure, weight, or energy expenditure. The trial was conducted between January 2013 and May 2016. The intervention was a 12-month protocol that consisted of 22 diet-specific group education sessions with registered dietitian nutritionists and clinical health educators. The study was single-blinded, as subjects could not be blinded to their diet assignment. All data were collected at baseline, 3, 6, and 12 months [17]. Of the 609 participants, 304 were randomized to the HLC arm. This secondary analysis included participants in the HLC arm with complete plasma lipid and diet data at baseline and 12 months (*n* = 208). 

Dietary intake data were collected via three, unannounced, 24-hour dietary recalls within a two-week window at each data collection time point. Food composition and energy data were collected using the Nutrition Data System for Research (NDSR) software from the University of Minnesota Nutrition Coordinating Center. A standardized, multiple-pass interview approach was used for dietary recalls [16]. Blood samples used for plasma lipid analysis were collected from participants after a fast of at least 10 h. Plasma lipid data from fasting blood samples were analyzed by a laboratory certified by the lipid standardization program of the National Heart, Lung, and Blood Institute of the Centers for Disease Control and Prevention (Krauss Lab, Children’s Hospital Oakland Research Institute, Oakland, CA, USA). The LDL-C concentrations (normal range < 200 mg/dL) were calculated using the Friedewald equation (note that only one participant at only one time point had a triglyceride measurement over 400 mg/dL). HDL-C and triglyceride levels were measured using standard methods [16]. 

Data collectors were blinded to diet assignment. Participant data were managed in Research Electronic Data Capture (RedCap)—an electronic data-capture tool hosted at Stanford University [18]. Participants provided informed consent as approved by the Stanford University Human Subjects Committee.

### 2.2. Statistical Analysis 

Descriptive statistics were used to summarize baseline diet and health characteristics, demographic information, and dietary cholesterol intake. Statistical tests (chi-square tests for categorical variables, Fisher’s exact test with Monte Carlo approximation used for cells with *n* < 5, one-way ANOVA for continuous variables) were also performed to assess differences in demographic and baseline characteristics between HLC participants included in this secondary analysis *(n =* 208) and those with missing data *(n* = 96) who were excluded. 

Least squares regression was used to compare 12-month change in dietary cholesterol intake (per 100 mg) with 12-month change in plasma LDL-C (mg/dL). Models were run both unadjusted and adjusted for age, sex, baseline dietary cholesterol intake, 12-month change in saturated fat intake, and 12-month weight change. While the regression analyses were done using continuous data, for ease of visual comparison 12-month changes in lipid variables (i.e., LDL-C, HDL-C, and triglycerides) are displayed in tables and figures by tertiles of 12-month change in dietary cholesterol intake (i.e., low change, moderate change, and high change). Correlations between 12-month change in dietary cholesterol intake and 12-month changes in LDL-C, HDL-C, and triglycerides were computed using Pearson’s correlation coefficients, and 12-month changes in lipids are displayed in box plots by tertile of dietary cholesterol intake. 

For comparison to the HLC analyses described above, all descriptive and primary analyses were repeated for the HLF group. Because the focus of this analysis is on the HLC group, and due to limitations of the number of tables and figures in the main paper, some of the HLF data and findings are only available in the supplemental materials. 

Significance levels for all analyses were set at an alpha of 0.05. All statistical analyses were performed using SAS University Edition (SAS Studio 3.6; SAS Institute Inc., Cary, NC, USA).

## 3. Results

### 3.1. Baseline Characteristics of the Study Population

Of the 304 participants randomized to the HLC arm (Figure A1), 208 (68%) had complete diet and lipid data at baseline and 12 months and were included in this complete case analysis. There were no significant differences between the HLC subjects included in the analysis *(n =* 208) and those excluded *(n =* 96) with regard to sex, race, education, or baseline diet, but those with missing data were younger (38 ± 6.9 years vs. 40 ± 6.6 years, mean ± standard deviation (SD), heavier (99 ± 15 kg vs. 95 ± 16 kg), and had lower mean LDL-C level (109 ± 25 mg/dL vs. 116 ± 26 mg/dL) (Table A1). 

Participants reported consuming a mean of 322 ± 173 (SD) mg of dietary cholesterol per day at baseline, with 103 participants (~50%) reporting consuming >300 mg/day. The following characteristics were not significantly different among the tertiles of 12-month change in dietary cholesterol intake: age, sex, race, education, baseline weight, and plasma lipids (Table 1). Baseline levels of cholesterol intake were not significantly different between the tertiles. Baseline carbohydrate intake was similar among the tertiles (*p* = 0.18). There were inverse relationships between 12-month change in dietary cholesterol intake and baseline levels of cholesterol (mg), total fat (g), saturated fat (g), protein (g), and calories (kcal; all *p* < 0.01). At baseline, the lowest, middle, and highest tertiles of 12-month dietary cholesterol change consumed 414 ± 200, 307 ± 148, and 245 ± 120 mg/day (mean ± SD), respectively (Table 1). Baseline diet and demographic data for the HLF arm are shown in Table A2.

### 3.2. Changes in Cholesterol Intake

At 12 months, mean intake of dietary cholesterol was 460 ± 227 (SD) mg/day in the HLC group—an increase of 42.9% from baseline—and most participants (76%) reported consuming >300 mg/day. From baseline to 12 months, mean dietary cholesterol for the total HLC group included in the analyses increased by 137 ± 206 mg/day. In comparison, the total HLF group decreased cholesterol consumption by 82 ± 161 and had a mean intake of 238 ± 128 at 12 months. In the HLC group, the lowest tertile had a 12-month mean change of −102 ± 131 mg/day, and the middle and highest tertiles increased by 112 ± 56 and 401 ± 185 mg/day, respectively (Table 2a,b). The interquartile ranges for 12-month change in dietary cholesterol intake (mg/day) were (−148.4, −20.0), (62.7, 153.5), and (274.1, 476.2) for the lowest, middle, and highest tertiles of the HLC arm, respectively. 

### 3.3. Correlations between Changes in Cholesterol Intake and Lipid Profile

The 12-month changes in cholesterol intake (Figure 1A) were not significantly correlated with changes in LDL-C (*r* = 0.04, *p* = 0.53), HDL-C (*r* = 0.1, *p* = 0.16), or triglycerides (*r* = −0.1, *p* = 0.06) for the HLC arm. Levels of LDL-C were relatively stable across all time points for all tertiles of the HLC arm (Figure 1B). Similar lack of significant correlations (LDL-C (*r* = 0.09, *p* = 0.18), HDL-C (*r* = 0.1, *p* = 0.15), triglycerides (*r* = 0.03, *p* = 0.69)) and LDL-C relative stability across time points for all tertiles were observed for the HLF arm (Figure 2).

The 12-month change in LDL-C for each 100 mg increase in daily dietary cholesterol intake in the HLC arm was not statistically significant in either the unadjusted model (0.37 mg/dL; 95% confidence interval (CI): −0.14 to 6.2) or the model adjusted for age, gender, baseline dietary cholesterol intake, and 12-month changes in saturated fat intake and weight (0.01 mg/dL; 95% CI: −1.4 to 1.5) (Table 3a,b). The 12-month changes in HDL-C and triglycerides per 100 mg increase in daily dietary cholesterol intake were also not statistically significant for either the HLC or the HLF arm. 

### 3.4. Qualitative Analysis of Sources of Dietary Cholesterol Increase

The study health educators that advised DIETFITS study participants throughout the trial indicated that many of those assigned to the HLC arm frequently consumed eggs for breakfast, substituting for previous habitual breakfast fare. Figure A2 shows the mean intake of eggs and refined grains for each tertile of 12-month change in dietary cholesterol intake at each time point in the study in the HLC arm. Mean intake was 0.86 ± 0.9 eggs/day at baseline and 1.45 ± 1.1 eggs/day at 12 months for all participants (*n* = 200, *p* < 0.001; egg data missing for 8 participants). The average egg contains approximately 207 mg of cholesterol, meaning participants’ mean dietary cholesterol intake from eggs was 178 mg and 315 mg at baseline and 12 months, respectively [19]. A secondary objective in this study was to explore the extent to which changes in dietary cholesterol intake were due to changes in egg consumption and whether changes in egg consumption were associated with changes in refined grain and added sugar intake. Participants in the highest tertile ate an average of one egg per day at baseline and two eggs per day at 12 months, while those in the lowest tertile ate approximately one egg per day at both time points. All tertiles reduced both sugar and refined grain consumption.

## 4. Discussion

In this secondary analysis of the DIETFITS weight loss trial, substantial 12-month changes in dietary cholesterol were not significantly associated with 12-month changes in LDL-C, HDL-C, or triglycerides among those assigned to follow a healthy, low-carbohydrate diet. The findings were similar before and after adjustment for age, gender, baseline dietary cholesterol intake, and 12-month changes in saturated fat intake and weight.

The one-third of the study population with the greatest 12-month change in mean dietary cholesterol intake increased from 245 mg/day to nearly 650 mg/day—more than double the 300 mg/day upper limit recommended in the 2010 and prior Dietary Guidelines for Americans. Notably, this portion of the study population also substantially increased average egg consumption from approximately one egg/day to two eggs/day. The calorie reduction in each tertile at each time point after baseline appears to be largely due to decreases in refined grains and added sugars, with some but not all of those calories being replaced by an increase in egg consumption for the middle and high tertiles. 

These results are supportive of the change in recent guidelines that eliminate recommending a specific upper cut point for cholesterol intake. From 1968 to 2015, both the American Heart Association (AHA) and the Dietary Guidelines for Americans (DGA) recommended no more than 300 mg/day of dietary cholesterol, stating that dietary cholesterol intake should be restricted due to its positive association with total and LDL cholesterol concentrations [2,3,5,6,7,20,21,22,23]. Consumption of eggs, especially egg yolks, was advised against for those aiming for heart health [2]. However, recent updates to the AHA and DGA guidelines have eliminated specific dietary cholesterol target recommendations, reflecting a shift away from emphasizing specific nutrients (e.g., cholesterol) and foods (e.g., eggs) toward a greater focus on heart-healthy dietary patterns such as healthy Mediterranean-style, DASH-style, and vegetarian-style diets [13].

This study’s findings align with the 2020 AHA Scientific Advisory position statement’s general recommendation to focus primarily on establishing healthy dietary patterns, rather than on trying to limit cholesterol intake. However, our findings are not consistent with the AHA’s specific recommendation regarding amount. The AHA recommendation is to limit egg intake to current levels, with healthy individuals being able to include a whole egg or equivalent daily. Our findings suggest a potentially modified recommendation on the basis of impact on LDL-C concentrations, i.e., up to two eggs/day. While the HLC group in the DIETFITS trial reported a wide range of 12-month changes in dietary cholesterol, no significant association with 12-month changes in LDL-C was observed. Mean LDL-C levels remained relatively stable for participants regardless of change in dietary cholesterol intake over 12 months. More specifically, egg intake also did not seem to be associated with LDL-C levels. The third of the study population that had the least change in dietary cholesterol intake consumed approximately one egg/day at both baseline and 12 months, while the third of the study population that saw the greatest increase in cholesterol intake also had the greatest increase in egg consumption, doubling mean intake from one egg/day at baseline to two eggs/day at 12 months. 

Notably, these findings occurred in the context of a weight loss study that involved shifts to an overall more healthful dietary pattern including substantial reductions in added sugars, refined grains, and overall energy intake. The AHA and DGA guidelines refer to dietary intake broadly and not specifically to intake in the setting of a weight loss diet. Regardless, it is clear that dietary cholesterol intake is not universally detrimental with regard to blood lipid levels and warrants further investigation in the settings of weight loss and weight neutral diets as well as among various dietary patterns.

In the landmark feeding studies of Hegsted and Keys from the 1950s and 1960s, saturated fat and dietary cholesterol were both determined to adversely affect plasma lipids, with saturated fat having the greatest impact [6,7,11]. However, Keys later published a review of the literature concluding that dietary cholesterol did not, in fact, play a major role in increasing serum cholesterol levels [24]. In line with Keys and the recent 2019 AHA Scientific Advisory on cholesterol, Carson et al. conducted a meta-analysis of 11 controlled studies and reported no significant association between dietary cholesterol and LDL-C [13]. Notably, the 11 studies selected from >50 randomized controlled trials [25] were all feeding studies, and all had similar polyunsaturated/saturated fatty acid ratios between comparison diets; the added rigor of these inclusion criteria provides a plausible explanation for differences between the Advisory conclusions and previous studies [6,7,11,25]. Our findings are aligned with the AHA Advisory report.

This study had several strengths. First, the study was conducted with a relatively large sample size over 12 months. Second, the study population included a weight range that was generalizable to much of the US population and had near equal representation of the genders. Additionally, there was extensive dietary data assessment using the gold standard for dietary data collection with 78% of participants completing 100% of the staff-administered, unannounced, multipass, 24-hour dietary recalls. Finally, study participants’ wide variability in 12-month change in dietary cholesterol intake increased the likelihood of identifying an association with LDL-C should an association exist. 

The study also had notable limitations. First, this was a secondary analysis from a randomized weight loss diet trial that was not designed to specifically test for causal effects of changes in dietary cholesterol intake on plasma lipids. Second, this analysis focused on only one of two intervention groups, thereby shifting to an observational design. Third, the use of tertiles of 12-month change in cholesterol for presentation purposes led to several important and potentially confounding differences at baseline between the tertiles (e.g., differential baseline cholesterol intake). However, this was addressed in the regression analysis which was done using continuous variables and adjusted for potential confounders. Complete data were available for only 68% of potential participants; fortunately, most baseline characteristics were similar between participants and those excluded due to missing data. Exceptions were age, baseline weight, and LDL-C. Despite using the most rigorous type of dietary data assessment tool, by nature of self-report, the data are subject to inaccuracies. In addition, energy (kcal) intake data were based on self-report, and therefore energy intake was likely underreported; however, this is a common, unfortunate but accepted aspect of nutritional trials. As we were looking at change in intake, we would expect underreporting to be consistent across time for participants and, therefore, not to affect our analyses to a large degree. Finally, as HLC-group participants’ egg consumption increased, both their refined grain and sugar consumption decreased. While this suggests that participants replaced some of the energy intake from carbohydrate-rich food sources with eggs, future studies should rigorously analyze the dynamics and impacts of such dietary shifts on LDL-C. 

In conclusion, the results of this secondary analysis suggest that physicians concerned about their patients’ blood lipid levels should take into consideration contextual factors in determining how to counsel them about dietary cholesterol intake, particularly the source of the cholesterol and the overall dietary quality pattern. Confusion or concern about dietary cholesterol might be avoided if the important inter-relationships of nutrients, foods, and food patterns are considered when advising patients on what to avoid and limit.

## Figures and Tables

**Figure 1 nutrients-13-01935-f001:**
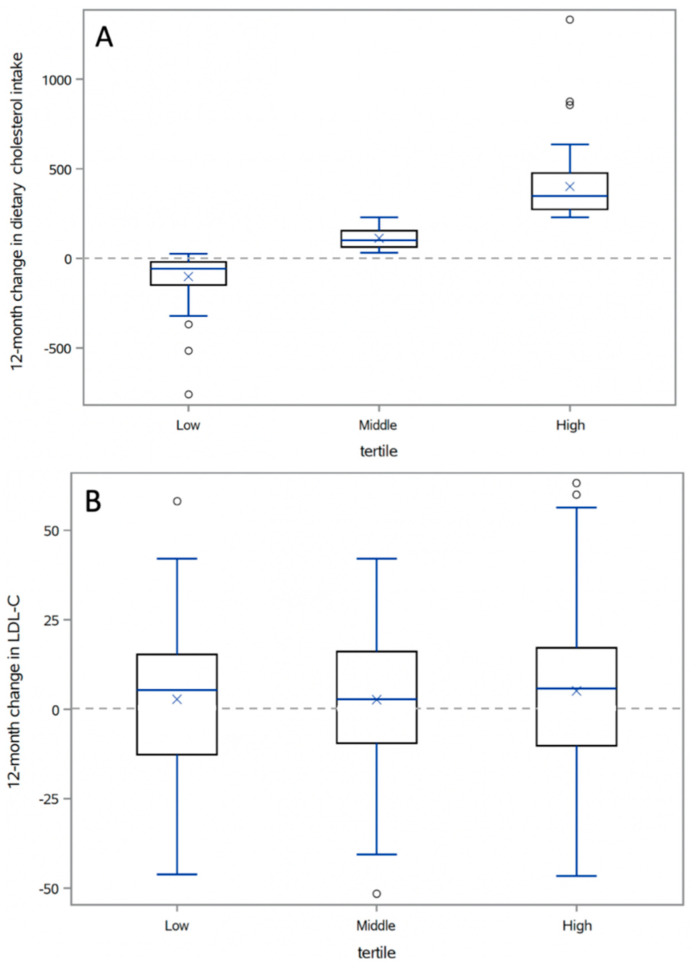
(**A**) Box plots of 12-month changes in dietary cholesterol intake by tertile of 12-month change in dietary cholesterol for healthy low-carb (HLC). Boxes represent interquartile range (IQR); center line in box is median, “*X*” within the box is the mean; upper and lower tails are 1.5*X* IQR. (**B**) Box plots of 12-month changes in LDL cholesterol intake by tertile of 12-month change in low-density lipoprotein (LDL) cholesterol for healthy low-carb (HLC). Boxes represent interquartile range (IQR); center line in box is median, “*X*” within the box is the mean; upper and lower tails are 1.5*X* IQR.

**Figure 2 nutrients-13-01935-f002:**
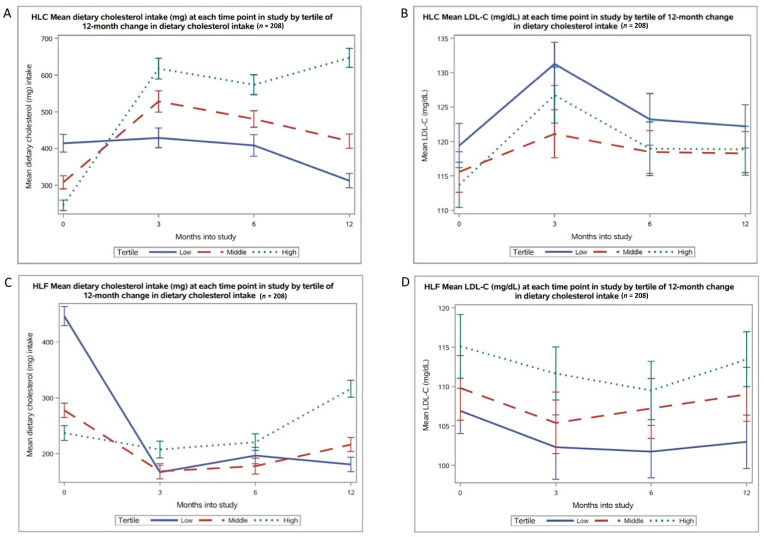
Levels of both dietary cholesterol intake and low-density lipoprotein cholesterol (LDL-C) (unadjusted for saturated fat or other factors) in the healthy low-carbohydrate (HLC) and healthy low-fat (HLF) arm at baseline, 3, 6, and 12 months: (**A**) HLC cholesterol; (**B**) HLC LDL-C; (**C**) HLF cholesterol; (**D**) HLF LDL-C.

**Table 1 nutrients-13-01935-t001:** Baseline demographics of participants by tertiles of 12-month change in dietary cholesterol intake (mg) ^1^ in the healthy low-carbohydrate arm (HLC).

		HLC-Tertile ^2^			
	Total *(n =* 208)	Lowest *(n* = 69)	Middle (*n* = 70)	Highest (*n* = 69)	*p*-Value ^3^
Age	40 ± 6.6	40 ± 6.6	40 ± 6.7	41 ± 6.3	0.45
Gender, *n* (%)					0.48
Female	125 (60)	45 (65)	42 (60)	38 (55)	
Male	83 (40)	24 (35)	28 (40)	31 (45)	
Race, *n* (%) ^4^					0.66
White	151 (73)	52 (75)	48 (70)	51 (74)	
Other	56 (27)	17 (25)	21 (30)	18 (26)	
Education, *n* (%)					<0.001 ^5^
High School	7 (3)	4 (6)	1 (1)	2 (3)	
College Graduate	118 (57)	39 (57)	39 (56)	40 (58)	
Postgrad Degree	83 (40)	26 (37)	30 (43)	27 (39)	
Body Weight (kg)	95 ± 16	95 ± 15	94 ± 16	94 ± 17	0.94
Baseline Diet					
Calories (kcal)	2198 ± 638	2338 ± 613 ^a^	2271 ± 702 ^a^	1985 ± 537 ^b^	0.002
Carbohydrates (g)	243 ± 73	252 ± 72	246 ± 78	231 ± 69	0.18
Fat (g)	91 ± 34	100 ± 32 ^a^	96 ± 36 ^a^	78 ± 28 ^b^	0.0002
Saturated Fat (g)	30 ± 13	34 ± 14 ^a^	32 ± 12 ^a^	25 ± 10 ^b^	<0.0001
Protein (g)	92 ± 29	101 ± 30 ^a^	92 ± 28 ^ab^	85 ± 26 ^b^	0.003
Cholesterol (mg)	322 ± 173	414 ± 200 ^a^	307 ± 148 ^b^	245 ± 120 ^c^	<0.0001
Lipids (mg/dL)					
LDL-C ^6^	116 ± 26	119 ± 27	115 ± 24	114 ± 27	0.43
HDL-C ^6^	50 ± 9	51 ± 8	51 ± 9	49 ± 9	0.54
Triglycerides	125 ± 105	128 ± 62	119 ± 53	131 ± 65	0.48

^1^Data are expressed as means ± standard deviation (SD) unless otherwise indicated; ^2^ HLC: healthy low-carbohydrate, tertiles are based on 12-month change in dietary cholesterol intake from baseline (mg/day); ^3^
*p*-values calculated by chi-squared tests for categorical variables and one-way ANOVA for continuous variables for the three tertile columns; significance level set at α = 0.05; ^4^ 207/208 participants reported their race/ethnicity; ^5^ Fisher’s exact test with Monte Carlo approximation used to calculate *p*-value for cells with less than 5; ^a,b,c^ for baseline characteristics with *p*-value < 0.05 across tertiles from one-way ANOVA, pairwise differences are indicated by superscripts; pairs with a shared superscript are not different as determined by unpaired *t-*test; ^6^ LDL-C: low-density lipoprotein cholesterol, HDL-C: high-density lipoprotein cholesterol.

**Table 2 nutrients-13-01935-t002:** (**a**) Mean changes from baseline in diet, weight, and lipids by tertile of 12-month change in dietary cholesterol intake for (a) the healthy low-carbohydrate arm. (**b**) Mean changes from baseline in diet, weight, and lipids by tertile of 12-month change in dietary cholesterol intake for the healthy low-fat arm (HLF).

(a)				
		HLC-Tertile		
	Total (*n* = 208)	Lowest (*n* = 69)	Middle (*n* = 70)	Highest (*n* = 69)
Total dietary cholesterol intake at 12 months (mg)	460 ± 227	312 ± 159	420 ± 160	647 ± 214
12-month change in dietary cholesterol intake (mg)	137 ± 206	-102 ±131	112 ± 56	401 ± 185
Range (min, max)	(−760, 1332)	(−760, 26)	(32, 230)	(230, 1332)
Calories (kcal)	−507 ± 617	−664 ± 584	−578 ± 643	−277 ± 559
Carbohydrates (g)	−112 ± 76	−104 ± 68	−104 ± 83	−128 ± 75
Saturated Fat (g)	−1.9 ± 14	−7.7 ± 14	−3.3 ± 13	5.4 ± 11
Body Weight (kg)	−6.3 ± 6.8	−4.4 ± 6.4	−6.5 ± 6.5	−8.0 ± 6.9
LDL-C (mg/dL)	3.5 ± 20	2.8 ± 20	2.7 ± 18	5.1 ± 23
HDL-C (mg/dL)	2.8 ± 6.5	2.2 ± 6.2	2.4 ± 5.8	3.9 ± 7.4
Triglycerides (mg/dL)	−25 ± 48	−17 ± 44	−24 ± 46	−35 ± 52
**(b)**				
		**HLF-Tertile**		
	**Total** **(*n* = 208)**	**Lowest** **(*n* = 69)**	**Middle** **(*n* = 70)**	**Highest** **(*n* = 69)**
Total dietary cholesterol intake at 12 months (mg)	238 ± 128	181 ± 108	216 ± 107	316 ± 127
12-month change in dietary cholesterol intake (mg)	−82 ± 161	−265 ±103	−61 ± 39	79 ± 74
Range (min, max)	(−655, 365)	(−655, −144)	(−144, −4.4)	(0, 365)
Calories (kcal)	−485 ± 627	−723 ± 700	−422 ± 593	−310 ± 509
Carbohydrates (g)	−36 ± 81	−38 ± 90	−37 ± 84	−33 ± 71
Saturated Fat (g)	−11 ± 12	−19 ± 12	−8.8 ± 11	−6.6 ± 10.2
Body Weight (kg)	−5.6 ± 7.3	−7.0 ± 7.6	−5.5 ± 8.1	−4.1 ± 5.7
LDL-C (mg/dL)	−2.1 ± 20	−3.9 ± 18	−0.8 ± 24	−1.6 ± 18
HDL-C (mg/dL)	0.2 ± 6.0	−0.5 ± 6.2	0 ± 6.0	1.0 ± 5.6
Triglycerides (mg/dL)	−11 ± 55	−14 ± 51	−13 ± 55	−6.1 ± 59

Data expressed as means ± standard deviation; positive change indicates an increase from baseline. HLC: healthy low-carbohydrate, HLF: healthy low-fat, LDL-C: low-density lipoprotein cholesterol, HDL-C: high-density lipoprotein cholesterol.

**Table 3 nutrients-13-01935-t003:** (**a**) Linear regression model estimates for 12-month changes (mg/dL) in LDL-C, HDL-C, and triglycerides associated with each 100 mg increase in daily dietary cholesterol intake among those assigned to follow a healthy low-carbohydrate diet (HLC). (**b**) Linear regression model estimates for 12-month changes (mg/dL) in LDL-C, HDL-C, and triglycerides associated with each 100 mg increase in daily dietary cholesterol intake among those assigned to follow a healthy low-fat diet (HLF).

(a)			
Outcome	Model #1 Unadjusted Estimate (95% CI)	Model #2 ^1^ Adjusted Estimate (95% CI)	Model #3 ^2^ Adjusted Estimate (95% CI)
LDL-C (mg/dL)	0.37 (−0.7 to 1.5)	0.30 (−1.0 to 1.6)	0.01 (−1.4 to 1.4)
HDL-C (mg/dL)	0.26 (−0.1 to 0.6)	0.12 (−0.3 to 0.6)	0.05 (−0.4 to 0.5)
Triglycerides (mg/dL)	−2.52 (−5.2 to 0.1)	−0.92 (−4.0 to 2.1)	0.06 (−3.1 to 3.3)
**(b)**			
**Outcome**	**Model #1** **Unadjusted Estimate (95% CI)**	**Model #2 ^1^** **Adjusted Estimate** **(95% CI)**	**Model #3 ^2^** **Adjusted Estimate** **(95% CI)**
LDL-C (mg/dL)	1.17 (−0.5 to 2.7)	1.40 (−1.0 to 3.8)	0.57 (−1.9 to 3.0)
HDL-C (mg/dL)	0.37 (−0.13 to 0.88)	0.24 (−0.5 to 0.9)	0.10 (−0.6 to 0.8)
Triglycerides (mg/dL)	0.95 (−3.7 to 5.65)	0.71 (−5.4 to 6.9)	0.61 (−5.8 to 7.0)

LDL-C: low-density lipoprotein cholesterol, HDL-C: high-density lipoprotein cholesterol, CI: confidence interval. ^1^ Adjusted for age, gender, baseline dietary cholesterol intake (mg/day), and 12-month change in weight (kg); ^2^ adjusted for age, gender, baseline dietary cholesterol intake (mg/day), and 12-month changes in both saturated fat intake (g) and weight (kg).

## Data Availability

Data described in the manuscript, code book, and analytic code will be made available upon request pending application and approval by the corresponding author.

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
