# Peer review of "Associations of Changes in Blood Lipid Concentrations with Changes in Dietary Cholesterol Intake in the Context of a Healthy Low-Carbohydrate Weight Loss Diet: A Secondary Analysis of the DIETFITS Trial"

_nutrients, 2021, doi:10.3390/nu13061935_

Round 1
Reviewer 1 Report
In the material and method section, I recommend that you should indicate that patients have been informed of the study and that they have signed a consent form.
With regard to the Kcal calculations, I also recommend that you add in the theoretical framework the programme or software that you have used to obtain the nutritional results.
Author Response
Attached as word doc

Reviewer 2 Report
see file added

Author Response
Attached as word doc

Reviewer 3 Report
The manuscript describes the association between the amount of cholesterol intake and the serum lipids concentration, in particular LDL.
The manuscript is well presented and describes all necessary data for an overall understanding of the topic.
Although the manuscript contains few limitations, the authors are aware and all discussed in the discussion.
Few improvements could be done.
- While describing the study design it would be good to describe the sore of data collcedted at each time point (briefly). That would avoid the need of searching the reference literature.
- I my opinion it would be better (if possible) to move Figure A3 from the appendix to the main body of the manuscript as it describe important information (the timeline) of the measurements.
Author Response
Attached as word doc

Reviewer 4 Report
The paper is valuable and adds to the need to assess dietary patterns from several perspectives. My main concern is the references. As I have indicated below, some are not appropriate to the text, some are superfluous and several need careful adjustment to the actual citation. Check consistency of using upper and lower case for headings and also check the online recommendations as some do not work. Specific comments follow.
Line 42
Reference 2 should be reference with reference 1, although either one of these is sufficient
References for limiting cholesterol would be 3-6, although again I don’t really see why so many references to this are needed.
Line 45
References7-11
Ref 7 – This paper by Keys, Anderson and Grande looks at differences when the fat content of the diet is changed. It does not look specifically at changing dietary cholesterol. Indeed, the article states on page 1 that “The low-fat-base diet provided an average of about400 mg less cholesterol daily than did the corresponding house diet, but variations of twice this amount of cholesterol are without effect on the serum-cholesterol.” Some years ago, I personally spoke to Ancel Keys about the fact that he was commonly quoted as the cause of the Dietary Guidelines for Americans recommending strict limits of dietary cholesterol . He was adamant that there was no relationship – and he had detailed this in his paper published in 1956. See Keys A, Anderson JT, Mickesen O, Adelson A, Fidanza F. Diet and serum cholesterol in man: Lack of effect of dietary cholesterol. J Nutr. 1956 May 10;59(1):39-56. doi: 10.1093/jn/59.1.39 (available at https://academic.oup.com/jn/article-abstract/59/1/39/4722525). In this paper he quotes 8 studies (admittedly some very small), and concluded “..that in adult men the serum cholesterol level is essentially independent of the cholesterol intake over the whole range of natural human diets. It is probable that infants, children and women are similar.” In view of this, you should remove ref 7 from this list as including it makes it sound as though Keys had pushed the idea that dietary cholesterol was a major risk factor for CVD. By removing this reference from here – and inserting it later (see comment for lines 45-47), you have the opportunity to correct the common accusation that Keys was somehow responsible for recommending a reduction in consumption of eggs.
Ref 8 – Hegsted did consider there was a relationship between dietary cholesterol and LDL-cholesterol (as we can see in ref 9) but as I can only access the first page of the ref 8 paper, could I suggest you check the whole paper to ensure it does discuss dietary cholesterol and not just the effects of dietary fat on LDL-cholesterol.
Ref 9 – No problems with this paper which is valid for the point you make in the current paper.
Ref 10 – Needs more detail as the general website is insufficient.
Ref 11 - This reference is about the strength and validity of the association between LDL cholesterol levels and risk of atherosclerotic cardiovascular disease. It is not about links between dietary cholesterol and blood concentrations of LDL-cholesterol. I suggest it is not a suitable (or necessary) reference here. Also the addition of https://www.narcis.nl/publication/RecordID/oai:pure.rug.nl:publications%2F813615e3-c9bd-4750-a7c2-8b6383656f3a. leads to the abstract of a paper in Dutch. Suggest you remove everything in ref 11.
Lines 46-47 - The references here are good and it would also be a good place to correct the common misconception about Ancel Keys and note that he too did not find that dietary cholesterol was a factor in raising LDL-cholesterol (I have included the Keys et al reference above, but your ref 7 also states the true Keys’ position. Either or both can be used to advantage here).
Line 61 - Ref 17 – Please check and correct the citation. The first author is KeyTJ, followed by Appleby PN..
Line 220 following – It would be useful to note the percentage of dietary cholesterol that came from eggs. In the highest tertile for dietary cholesterol, you state this group averaged 2 eggs/day. As 2 eggs contains approx. 400 mg cholesterol, the highest tertile group also consumed considerably more of other sources of dietary cholesterol than the lowest tertile who averaged only 1 egg/day. This surely deserves some mention, here or in the discussion, even if only briefly.
Line 252 – I’d advise adjusting these references in keeping with the previous comments. Also I am not convinced that ref 22 is relevant (or needed) for this particular spot.
Line 253 – You might note in your commentary that in references 23 and 24, results were obtained from very small samples: 16 in the Roberts et al study and 17 in the Sacks et al study. Ref 3 is an excellent example of the thinking of the time from the AHA and the Dietary Guidelines for Americans and is probably all you need to make your point.
Line 264 – This is where it would be useful to discuss how much of the dietary cholesterol came from eggs.
Line 267 – It would be useful to note the very large variations in individuals’ dietary cholesterol consumption within each tertile.
Line 281 – 1950s and 1960s (no apostrophe). As noted earlier, Keys specifically stated, and quoted a list of studies to prove his point that dietary cholesterol was not the culprit. Keys A, Anderson JT, Mickesen O, Adelson A, Fidanza F. Diet and serum cholesterol in man: Lack of effect of dietary cholesterol. J Nutr. 1956 May 10;59(1):39-56. doi: 10.1093/jn/59.1.39.
Line 283 – in view of Keys’ findings and conclusions, it would be better to omit ‘however’ and just start by saying.. As Keys had noted (give ref as suggested), in the 2019 AHA …
Line 289 – another instance where reference 7 should be omitted.
Author Response
Attached as word doc

Round 2
Reviewer 2 Report
see enclosure

Author Response
Attached word document.